# Marine Power on Cancer: Drugs, Lead Compounds, and Mechanisms

**DOI:** 10.3390/md19090488

**Published:** 2021-08-27

**Authors:** Lichuan Wu, Ke Ye, Sheng Jiang, Guangbiao Zhou

**Affiliations:** 1Medical College, Guangxi University, Nanning 530004, China; richard_wu@gxu.edu.cn; 2School of Pharmacy, China Pharmaceutical University, Nanjing 211198, China; yekecpu@163.com; 3State Key Laboratory of Molecular Oncology, National Cancer Center/National Clinical Research Center for Cancer/Cancer Hospital, Chinese Academy of Medical Sciences and Peking Union Medical College, Beijing 100021, China

**Keywords:** marine environment drug, natural compounds, anticancer, mechanisms of action

## Abstract

Worldwide, 19.3 million new cancer cases and almost 10.0 million cancer deaths occur each year. Recently, much attention has been paid to the ocean, the largest biosphere of the earth that harbors a great many different organisms and natural products, to identify novel drugs and drug candidates to fight against malignant neoplasms. The marine compounds show potent anticancer activity in vitro and in vivo, and relatively few drugs have been approved by the U.S. Food and Drug Administration for the treatment of metastatic malignant lymphoma, breast cancer, or Hodgkin′s disease. This review provides a summary of the anticancer effects and mechanisms of action of selected marine compounds, including cytarabine, eribulin, marizomib, plitidepsin, trabectedin, zalypsis, adcetris, and OKI-179. The future development of anticancer marine drugs requires innovative biochemical biology approaches and introduction of novel therapeutic targets, as well as efficient isolation and synthesis of marine-derived natural compounds and derivatives.

## 1. Introduction

With its great biodiversity and unique chemical diversity, the marine environment is a huge treasury of medicinal resources [1,2,3]. There are 17 clinically available drugs that were based on marine natural products or their derivatives, and 28 drugs that are currently being tested in phase I–III clinical trials [4,5]. Cancer has been the second leading cause of mortality worldwide, causing approximately 10 million deaths in 2020 [6]. In the past decades, substantial efforts have been made to unveil the pathogenesis of cancers and new anticancer drugs have been developed [7,8,9,10]. Here, we review the original resources, anticancer effects, mechanisms, and clinical applications of selected marine-derived compounds including cytarabine (the first approved marine-derived anticancer drug), eribulin (a microtubule-depolymerizing drug), marizomib (a proteasome inhibitor), plitidepsin (a DNA synthesis inhibitor), trabectedin (a nucleotide drug), adcetris (an antibody drug conjugate), zalypsis (a new DNA binding alkaloid), and the largazole analogue OKI-179 (an HDAC inhibitor) (Table 1).

## 2. Cytarabine/Ara-C

Cytarabine (Figure 1), based on a lead compound from marine sponge was synthesized in 1959 by Walwick at the University of California, Berkeley. Cytarabine was the first clinically used marine anticancer drug to be approved in 1969 by the US Food and Drug Administration (FDA), mainly for treating acute leukemia [11,12]. Cytarabine is a cytosine base-containing pyrimidine nucleoside. It is converted from 1-β-d-arabinofuranosyluracil through the acylation of its hydroxyl group and conversion of the 4-carbonyl group in the pyrimidine ring into a thiocarbonyl group. Subsequently, the thiol group was replaced by an amino group and the acetyl group was hydrolyzed to obtain the final pyrimidine structure. The difference between the synthetic cytarabine and the naturally occurring cytarabine is that in the latter, there is a hydroxyl group in the 2’-β configuration of the sugar moiety. The resulting arabinose binds to the replicating DNA chain and inhibits the initiation and extension of the chain, resulting in the production of erroneous DNA.

Cytarabine is metabolized in vivo and transformed into its active form “ara-CTP”. The entire process involves three steps [13]. First, cytarabine is converted to the inactive form arabinouridine (Ara-U) after deamination and only small amount of cytarabine subsequently enters into the cell. Cytarabine is catalyzed into Ara-CMP via deoxycytidine kinase (DCK) in the cytoplasm. Second, Ara-CMP is transformed into Ara-CDP by pyrimidine nucleoside kinase. Third, the Ara-CDP is metabolized into Ara-CTP. Ara-CTP may be incorporated into the nucleotide chain of ribonucleic acid (DNA), which prevents chain elongation and causes chain breaks. By affecting the replication of the chain, the drug inhibits the proliferation of tumor cells. Studies have shown that Ara-CTP can bind to the topoisomerase complex limiting its activity by inhibiting DNA synthesis (Figure 1) [14]. 

The anticancer effect of cytarabine depends on the dosage. When cytarabine at low dosage enters into the structure of oligonucleotides, it reduces the binding capacity of transcription factors with their respective DNA binding elements and the transduction of new messenger RNA. At high doses, it exhibits a significant antitumor effect by inducing cell cycle arrest at the G0 and G1 phases. While cytarabine at medium and small doses can induce S phase cell cycle arrest. DNA fragmentation analysis shows that cytarabine induces cellular apoptosis in a dose-dependent manner (Figure 1) [15].

In the clinic, high and medium dosages of cytarabine are usually applied to treat acute myeloid leukemia (AML) [15]. However, serious side effects, such as bone marrow suppression, which will lead to anemia, leukopenia, thrombocytopenia, infection, and musculoskeletal and connective tissue abnormalities have limited its clinical use [16,17,18]. Therefore, combination medication becomes an alternative which can avoid serious adverse effects. Although decitabine combined with low-dose of Ara-C has a higher risk of bone marrow suppression and infection in the initial stage of treatment, patients can tolerate it and with the extension of the treatment course, the adverse reactions gradually decreased [19]. 

Drug resistance has always been a major problem in the clinical application of cytarabine. In recent years, research on drug resistance of cytarabine has gradually increased and studies have shown that drug resistance is related to biological factors, proteins, and gene mutations [20,21]. Cytarabine induces apoptosis of HL60 cells and activates nuclear factor-κB (NF-κB), which may cause early resistance of leukemia cells to cytarabine. It has been reported that NF-κB plays an important role in anti-apoptosis [22,23,24,25]. Deoxycytidine kinase (DCK) protein, one of the most critical proteins identified in research reports, is the key rate-limiting enzyme for the phosphorylation of cytarabine in the body. It has been found that DCK is down-regulated and mutated in drug-resistant mouse AML cell lines. As a result, the first phosphorylation of cytarabine in the body is restricted, affecting the activation and the therapeutic effect of cytarabine in the body [26].

## 3. Eribulin/E7389

Eribulin (Figure 2) is a derivative of halichondrin B which was isolated from *Halichondria okadai* [27,28]. Eribulin was approved by FDA for treatment of breast cancer and liposarcoma in 2010 and 2016, respectively [29]. The synthesis of eribulin is complicated and difficult [30]. Eribulin is a microtubule-depolymerizing drug, which interferes with the mitotic phase of cell cycle. Eribulin mainly binds with high affinity to the ends of positive microtubules, preventing the polymerization of tubulin. Eribulin can also bind to soluble α- or β-tubulin, decreasing the effectiveness of subunit polymerization and inducing cell cycle arrest at the G2/M phase and apoptosis through mitochondrial obstruction (Figure 2) [31]. In preclinical studies, eribulin has shown antitumor activity in various types of cancer, including colon cancer, glioblastoma, head and neck cancer, melanoma, non-small cell lung cancer (NSCLC), ovarian cancer, pancreatic cancer, and small cell lung cancer [32].

Eribulin can inhibit tumor metastasis by inhibiting epithelial-mesenchymal transition (EMT) and inducing mesenchymal-epithelial transition (MET). Yoshida et al. found that treatment of triple negative breast cancer (TNBC) cells with eribulin led to MET [33]. Eribulin treatment decreases expression of mesenchymal marker genes and increases the expression of epithelial markers. Funahashi et al. reported that eribulin remodels tumor vasculature in human breast cancer xenograft models by increasing microvessel density and decreasing mean vascular areas and branched vessels in tumor tissues [34]. Further, the authors reported that eribulin affects the expression of genes involved in the angiogenesis signaling pathway and the EMT pathway, and was posited to be related to the decrease in hypoxia. Clinical studies support the use of eribulin in the treatment of advanced or metastatic breast cancer, and also indicate that eribulin may treat solid tumors that are resistant to other types of microtubule kinetic inhibitors. In addition, the combination of eribulin and paclitaxel exerts a synergistic anti-proliferative effect on TNBC cells by increasing the expression of E-cadherin and decreasing the expression of cell mesenchymal markers [35]. 

## 4. Marizomib/NPI-0052

Proteasome degradation plays a critical role in the survival of malignant tumor cells. The increase of proteasome has been observed in various types of tumor cells, indicating that the survival and growth of tumor cells rely on the proteasome [36]. In the late stage of plasma cell differentiation, proteasome activity decreases with the accumulation of immunoglobulin, leading to the accumulation of ubiquitinated protein [37]. Marizomib (NPI-0052), a natural compound isolated from marine *Salinispora tropica* [38], possessing a γ-lactam-β-lactone fragment is a proteasome inhibitor. Studies have shown that marizomib induces loss of mitochondrial membrane potential, increases ROS production, cytochrome C/Smac release and activation of caspases, leading to multiple myeloma cell apoptosis [39]. Marizomib may also exert antitumor activity through the inner mitochondrial apoptotic pathway and death receptor pathway (Figure 3) [40,41,42]. Inhibition of the proteasome by marizomib is generally considered to be the result of inhibiting the NF-κB pathway. Proteasome inhibition increases the level of IκBα, thereby inhibiting the NF-κB pathway, resulting in a decrease in the production of anti-apoptotic factors, angiogenic factors and inhibition of apoptosis [43]. Proteasome inhibition leads to an imbalance in the self-regulation of cyclins, cyclin-dependent kinases and other cell cycle regulatory proteins and this can disrupt cell division. 

Chauhan et al. found that intravenous injection of single or multiple doses of marizomib (0.15 mg/kg) into mice can inhibit proteasome activity in peripheral organs, but not in the brain [39]. Marizomib is rapidly distributed to various organs and tumors from the vascular compartment, and clearly inhibits the caspase-like, trypsin-like, and chymotrypsin-like activity of proteasomes. Marizomib synergizes with the immunosuppressive agent pomalidomide in treatment of multiple myeloma by inducing apoptosis by activation of caspases and downregulation of CRBN, MYC, IRF4, and MCL1 [44]. 

Clinical studies have shown that Marizomib is effective in treating neoplastic malignancies including multiple myeloma, brain tumor, and glioma, and was approved by FDA in 2013 and the European Medicines Agency (EMA) in 2014 for treatment of multiple myeloma [45,46,47]. Marizomib is well tolerated, and adverse events include nausea, neutropenia, pneumonia, anemia, thrombocytopenia, and febrile neutropenia. Marizomib also causes central nervous system toxicity, such as cognitive changes, expressive aphasia, visual or auditory hallucinations, disorientation, and unstable gait, which can be relieved after ceasing the drug [48].

## 5. Plitidepsin/Aplidine

Plitidepsin (aplidine), a natural compound isolated from *Aplidium albican* [49], is a cyclic depsipeptide whose structure is similar to that of bisbenzophenone [50]. Plitidepsin has strong antitumor activity with low toxicity and was approved by FDA in 2004 to treat multiple myeloma [51]. 

The mechanism of plitidepsin in treating multiple tumors has been thoroughly investigated. Plitidepsin exhibits anticancer effects by inhibiting tumor cell proliferation and inducing cell apoptosis in multiple myeloma (MM), plasmacytoma, prostate cancer, pancreatic cancer, and ovarian cancer [52,53]. Plitidepsin treatment resulted in oxidative stress, rapid activation of Rac1 GTPase, and continuous activation of c-Jun N-terminal kinase (JNK) and p38 mitogen-activated protein kinase (p38/MAPK), leading to caspase activation and cell apoptosis (Figure 4) [54]. Recent studies have found that the translation elongation factor eEF1A2 is the primary target of plitidepsin, which can bind to eEF1A2 at the interface between domains 1 and 2 of this protein in the GTP conformation [51,55,56]. It has been further reported that plitidepsin also inhibits the activation and subsequent destruction of aggregates in lysosomes, which accumulate excessive misfolded proteins in cells, thereby triggering apoptosis. In chronic lymphocytic leukemia (CLL), plitidepsin affects the viability of leukemic cells through direct and indirect pathways. It inhibits the malignant B-CLL clone directly and restrains tumor growth by indirect modification of the micro-environment [53]. A recent study showed that plitidepsin analogs PM01215 and PM02781 inhibit angiogenesis in vitro and in vivo [57].

## 6. Trabectedin/ET-743

Trabectedin (ET-743) is a tetrahydroisoquinoline alkaloid derivative which was approved to treat liposarcoma and leiomyosarcoma by EMA and FDA in 2007 and 2015, respectively [58,59]. Trabectedin is isolated from extracts of *Ecteinascidia turbinata* and described as able to prevent oncogenic transcription factors from binding to the target promoters [60]. Trabectedin contains three fused tetrahydroisoquinoline rings (A, B, and C rings) [61]. The A and B rings form a rigid five-ring skeleton, which is connected to the C ring through a 10-membered lactone bridge. Ring A and B insert into deoxyribonucleic acid (DNA) by interacting with DNA minor grooves through hydrogen bonds and Van der Waals forces to promote the alkylation of adjacent nucleotides in the same or opposite strands of DNA [62]. The N2 of guanine in the 5′-CGG, TGG, GGC, AGC sequence is the alkylation site [63]. In addition, protonated amines are involved in the production of reactive imine ions (N2) that catalyze DNA binding [64]. The carbon ring protrudes from the deoxyribonucleic acid backbone and interferes with the deoxyribonucleic acid binding protein, while traditional alkylating agent drugs usually bind at the N-terminal or O-terminal. In addition, trabectedin inhibits transcription by binding to transcription RNA polymerase II (Pol II) and blocking its activity [65], leading to degradation of Pol II enzyme (Figure 5) [66]. 

Studies in vivo and in vitro have shown that trabectedin can reduce tumor growth and regulate the tumor micro-environment. In addition to the approved treatment of soft tissue sarcoma and ovarian cancer, trabectedin is undergoing clinical evaluation with other cancer types, including breast cancer, bone cancer, and prostate cancer [60]. Trabectedin inhibits transcription by reducing the transactivation ability of the chimeric protein, achieving some success in myxoid liposarcoma and Ewing′s sarcoma and other sarcoma subtypes, and ultimately leading to adipocyte differentiation and apoptosis [67]. The clinical application of trabectedin was first applied to soft tissue sarcoma (STS). A number of clinical phase I studies have found that it has a good effect on ovarian cancer when used alone or in combination with other chemotherapeutic drugs, such as cisplatin [68,69,70]. Phase II and III clinical studies have shown that trabectedin alone or in combination with other anticancer drugs is effective in soft tissue tumors, especially liposarcoma and leiomyosarcoma, and prolongs the survival time of the patients [30,71,72,73].

Clinical application of trabectedin is gradually expanding. It is reported that trabectedin could regulate the tumor micro-environment (TME) by decreasing the concentration of immune cells, such as monocytes and tumor-associated macrophages, and indirectly reducing the production of inflammatory mediators, such as interleukin-6, interleukin-8 and vascular endothelial growth factor, and affecting the expression of extracellular matrix-related genes [74]. Trabectedin could also inhibit genes and pathways related to the phenotype of cancer stem cells [75,76]. Moreover, trabectedin-induced DNA damage in an RNA-DNA hybrid-dependent manner in Hela cells has been observed. This RNA-DNA hybrid is also called R-loop, and high level of R-loops sensitizes tumor cells to trabectedin. [77].

## 7. Adcetris/Brentuximab Vedotin

Adcetris, or Brentuximab Vedotin is an antibody drug conjugate which couples a CD30 antibody and a tubulin targeted compound, monomethyl auristatin E (MMAE). Compound MMAE is a derivative of natural compound dolastatin 10 isolated from *Dolabella auricularia* [78,79,80]. Adcetris was approved in 2011 to treat Hodgkin’s lymphoma (HL) and systemic anaplastic large cell lymphoma (ALCL) [81]. Adcetris consists of three parts: a monoclonal antibody with a targeting function, which can guide adcetris to its target cells; an effective cytotoxic compound called the payload or warhead, which has the pharmacological effect of killing tumor cells; and a central part composed of hexanoic acid and maleimide that has a linking function. It can covalently connect the first two parts and be further responsible for releasing payloads and linkers in the target cells [82,83,84,85]. Brentuximab is a monoclonal antibody that locates tumor cells by searching for the tumor marker CD30 membrane protein. Vedotin is the second part of the conjugate, including the linker and the cytotoxic principle MMAE. To meet the requirements, the main ingredient of linker in adcetris is cathepsin-cleavable valine-citrulline, which can be cut precisely to release a cytotoxic compound. Adcetris exerts its anticancer activity in a few steps in vivo. First, the recognition of brentuximab antibody by CD30 protein leads to the decomposition of adcetris. Then, the selective proteolytic cleavage of adcetris promoted by lysosomal cysteine protease releases vedotin. Second, vedotin releases MMAE in cytoplasm. When free MMAE reaches its goal, it prevents cell division in G2/M by disrupting microtubule dynamics, and ultimately induces cell death (Figure 6) [85,86].

Adcetris is used in the clinic as a consolidation therapy after autologous stem cell transplantation (ASCT). Some studies have evaluated the efficacy of adcetris as a consolidation therapy for adult patients with Hodgkin′s lymphoma at risk of relapse or progression after receiving ASCT through a randomized, double-blind, Phase 3 trial [87]. During the 30-month follow-up, in the main independent retrospective analysis, treatment with adcetris, compared with placebo, significantly prolonged progression-free survival (PFS; 42.9 months vs. 24.1 months). Adcetris has a certain effect on other CD30-positive lymphomas. The efficacy of adcetris on B-cell lymphoma was evaluated in a phase II trial. The study included 65 patients with CD30-positive relapsed or refractory B-cell lymphoma who were given 1.8 mg/kg of adcetris. Among 48 evaluable basal cell carcinoma patients, eight and thirteen patients achieved complete and partial remission, respectively [88].

Adcetris has attracted great attention because of its excellent antitumor activity and its acceptable side effects. The successful development of adcetris indicates that CD30 can be used as a therapeutic target for HL and ALCL, which leads to the development of more CD30-targeting agents. Currently, more than 80 active phase I–III clinical trials have been carried out to evaluate the efficacies of adcetris in cancers, but drug resistance has been reported in patients treated with adcetris. Strategies to overcome drug resistance and combinatory regimens to achieve maximal efficacy are being tested in lymphoma [89].

## 8. Zalypsis/M00104

Zalypsis, an alkaloid newly isolated from *Joruna funebris*, is a tetrahydroisoquinolone alkaloid that is structurally similar to ascidianin and nephromycin [90]. It binds covalently to guanine residues in for example, the AGG, GGC, AGC, CGG, and TGG groups to form a DNA-zalypsis adduct, leading to early transcription inhibition and a double-stranded DNA break, affecting cell proliferation and development (Figure 7) [91,92]. Zalypsis shows antitumor activity against cell lines of bladder cancer, gastric cancer, liver cancer, prostate cancer, pancreatic cancer, thyroid cancer, sarcoma, leukemia, and lymphoma [91,93,94,95,96]. In multiple myeloma, zalypsis in combination with bortezomib and dexamethasone exerts enhanced efficacy with acceptable toxicity in vitro and in vivo [97]. Phase II and Phase III clinical trials have shown that zalypsis is effective in treating Ewing’s sarcoma, urothelial cancer, multiple myeloma, endometrial cancer, and cervical cancer [91,98].

## 9. Largazole and Its Analogues

HDACs are responsible for the deacetylation of lysine residues in histone and non-histone proteins, leading to chromatin condensation and transcriptional repression [99]. To date, eighteen human HDACs have been identified, and they are divided into four classes (I–IV). HDACs, especially class I HDACs, are overexpressed in various cancer cell types [100,101] and contribute to tumor development and progression [102,103]. Thus, HDACs have been validated as valuable anticancer targets, and four HDAC inhibitors, including vorinostat, romidepsin, belinostat, and panobinostat, have been approved by FDA for the treatment of hematologic malignancies including T-cell lymphoma and multiple myeloma. Selective HDAC inhibitors are highly desirable since they can not only serve as pharmacological tools to help elucidate the biological roles of specific HDAC class or isoforms, but also as therapeutic agents with potentially reduced side effects commonly existing in the pan-HDAC inhibitors [104,105].

Largazole, **1**, a cyclic depsipeptide, was isolated from marine *cyanobacterium Symploca sp.* in 2008 by Luesch et al. [106]. This research group firstly revealed the robust potency of largazole against HDAC1 in the low nanomolar range and with selectivity over HDAC6 [107,108]. Subsequently, largazole was found to strongly inhibit HDAC1, HDAC2 and HDAC3 in the picomolar range in a biological assay developed by Williams et al. [109]. Studies suggests that largazole possesses a wide range of pharmacological activities, such as anti-inflammation [110], anti-liver fibrosis [111], anti-virus [112], osteogenic activity [113], and anticancer activity [114]. Largazole has received much attention due to its potent anti-proliferative activity against various cancer cells and substantial potency as a class I histone deacetylase (HDAC1) inhibitor. Wu et al. reported that largazole could potently and selectively suppress lung cancer cell proliferation and colony formation activity with no obvious cytotoxicity against normal bronchial epithelial cells [115]. Largazole induce a G1 cell cycle arrest via upregulation of p21 and down-regulation of E2F1. Liu et al. found that largazole showed stronger inhibition of colon cancer cell lines than of other type of cancer cells by screening the National Cancer Institute’s 60 cancer cell lines [116]. Largazole exerted its anti-colon cancer effects by inducing cell cycle arrest, cell apoptosis, and down-regulation of insulin receptor substrate 1 in vitro and in vivo [116]. Law et al. reported that largazole and glucocorticoid dexamethasone cooperate to induce E-cadherin (E-cad) localization to the plasma membrane of triple-negative breast cancer cells, leading to inhibition of cell invasion [117]. Michelle et al. designed a multidimensional screening platform and found largazole was an inhibitor of oncogenic Kirsten ras (KRAS) and the hypoxia inducible factor (HIF) pathway which could inhibit colon cancer proliferation and angiogenesis [118]. Gilson et al. performed RNA sequencing to identify altered gene expression upon largazole treatment [119]. Their results demonstrated that largazole could particularly decrease RNA polymerase II accumulation at super-enhancers and inhibit oncogene activities in cancer cells induced by super-enhancers. Recently, largazole was also found to inhibit proliferation of glioblastoma multiforme (GBM) cell lines in vitro at nanomolar concentrations [120]. More importantly, it was shown to be highly brain-penetrant and, therefore, could achieve therapeutically relevant doses in mouse brain which led to upregulation of neuroprotective genes, including Bdnf and Pax6. Wang et al. revealed that largazole markedly suppressed cell proliferation and induced apoptosis in non-small cell lung cancer (NSCLC) and chronic myeloid leukemia (CML) by decreasing the expression of oncogenic protein Musashi-2 (MSI2) [121]. The anticancer mechanisms of largazole are summarized in Figure 8.

These excellent properties of largazole have prompted extensive structural modifications and structure-activity relationship (SAR) studies with the aim of searching for more potent and selective HDAC inhibitors [122,123,124]. Structurally, largazole is a prodrug which can be converted to the free thiol form (**2**) through removal of the octanoyl group by esterases or lipases [109]. As is shown in the co-crystal structure of HDAC8 complexed with largazole thiol (**2**), the free sulfhydryl group of **2** as a warhead chelates Zn^2+^ in the active site of the HDACs [125]. The octanoyl group makes largazole more cell-permeant and allows facile liberation of the free thiol within the cell [109]. The 16-membered depsipeptide ring system acts as a surface recognition group which interacts with the surrounding hydrophobic residues on the outer rim of the enzyme, contributing to enzyme potency and selectivity. The four-atom length of linker connecting the Zn^2+^-binding group (ZBG) to the depsipeptide ring occupies the narrow hydrophobic channel [125]. The structural modifications of largazole are mainly focused on the linker, ZBG and depsipeptide ring system (mainly focusing on the L-valine residue and the 4-methylthiazoline–thiazole subunit).

Initial SAR studies suggested that largazole analogues with a four-atom linker, a *trans*-form alkene attached to the macrocyclic ring and an *S*-configured C-17 are critical for HDAC inhibitory activity and antiproliferative effect against cancer cells [108,126,127]. Introduction of an aromatic group in the linker region results in complete loss of HDAC inhibitory activity [128]. In order to fully ascertain the role of the metal-binding domain in HDAC inhibition, a series of largazole analogues with various ZBGs were prepared. Analogues **3** and **4** with an α-thioamide or mercaptosulfide, respectively, and analogues **5** and **6** which added a second heteroatom for multiple heteroatomic chelation with Zn^2+^ showed reduced inhibitory activity (Figure 9). However, these compounds conferred selectivity inhibition on HDAC6 or HDAC10 over HDAC1 [129,130,131]. In addition, our group identified an analogue **7** with an octyldisulfide side chain which confers a significant selectivity on HDAC1 over HDAC7 [132]. Together, these studies indicated that ZBGs have profound effects on HDAC inhibitory potency and isoform selectivity.

Many largazole analogues with a modified L-valine unit have been reported. Different amino acid residues, such as Ala, Gly, Phe, Tyr, Asp, His, Leu, 1-naphthylmethylglycine, and 1-allylglycine were introduced into largazole at the C2 position and these modifications were well tolerated, albeit with a minimal loss in inhibitory activity or isoform selectivity [108,127,128,133,134]. Recently, Lei et al. reported a promising analogue (**8**) (Figure 10) that had an S-Me L-cysteine (MeS) substitution at the C2 position [135]. This compound showed a more potent inhibitory activity on HDAC1 than largazole thiol (**2**), and exhibited a comparable selectivity for HDAC1 over HDAC6.

A few largazole analogues with modified thiazole-thiazoline groups have been reported. The elimination or replacement of the methyl group of 4-methylthiazoline with an ethyl or a benzyl group is tolerated, indicating that 4-methylthiazoline moiety is not necessary for HDAC inhibition [136,137]. Based on this assumption, our group reported the largazole analogue (**9**) with a more hydrophobic tetrazole group in place of the 4-methylthiazoline, which showed potent inhibition of HDAC1, HDAC2, and HDAC3 (HDAC1, IC_50_ = 100 nM; HDAC2, IC_50_ = 224 nM; and HDAC3, IC_50_ = 31 nM) and better selectivity for HDAC1 over HDAC9 than largazole [138]. Based on the largazole analogue (**10**) discovered by our group, which contains an unsaturated dehydrobutyrine (Dhb) in place of 4-methylthiazoline, we replaced the L-valine region of **10** with different amino acid residues, such as glycine, butyrine, leucine, t-butylglycine and methionine [139,140]. Among these, compound **11** with L-alanine in place of L-valine showed the most potent and selective inhibitory effects on HDAC1-3 (HDAC1, IC_50_ = 1.3 nM; HDAC2, IC_50_ = 1.6 nM; and HDAC3, IC_50_ = 3.2 nM), and promising anti-proliferative activity against cancer cell lines Molt-4 and A549 (Molt-4, GI_50_ = 2.1 nM; A549, GI_50_ = 7.5 nM). This compound showed desirable antitumor efficacy in a PC3 xenograft model with no apparent toxicity [140]. 

Analogue **12** was obtained by replacing the 4-methylthiazoline group with a simplified α-aminoisobutyric acid residue, which showed a 20-fold reduced nanomolar HDAC inhibition compared to largazole thiol [133]. Recently, Nan and coworkers identified a bisthiazole-based HDAC inhibitor (**13**) through structural simplification of largazole skeleton. This compound exhibited potent nanomolar HDAC inhibition (HDAC1, IC_50_ = 3.64 nM; HDAC2, IC_50_ = 3.02 nM; and HDAC3, IC_50_ = 4.96 nM) and strong antiproliferative activity against NCI-N87 (IC_50_ = 28 nM) and T47D cells (IC_50_ = 40 nM). Furthermore, **13** showed good antitumor efficacy in the HT-29 xenograft model with good oral bioavailability and safety profile [124,141].

Among these largazole analogues, a compound named OKI-179 was identified as a candidate drug which has entered phase I clinical study to treat advanced solid tumors [142,143]. OKI-179 was discovered by Liu et al. who were aiming to improve the physiochemical properties and simplify the synthesis of largazole [144]. The results of in vitro enzyme activities assay indicated that OKI-179 displays significant inhibition towards class I, IIb, and IV HDACs. Wang et al. reported that compound OKI-179 could sensitize PD1 blockade-resistant B cell lymphomas to PD1 antibody treatment [145]. Treatment with OKI-179 leads to cell-cycle arrest, apoptosis, and growth inhibition in tumors. According to the mechanism of action of OKI-179, MHC proteins are downregulated in PD1 blockade-resistant B cell lymphomas (G1XP lymphomas). Upon OKI-179 treatment, MHCs including class I and II are up-regulated. Furthermore, MHC knockout attenuates OKI-179 induced tumor growth inhibition. OKI-179 and OKI-179/anti-PD1 treatment activates tumor-infiltrating lymphocytes (TILs) in G1XP lymphomas. These results displayed that OKI-179 sensitizes lymphomas to PD1-blockade by enhancing tumor immunogenicity.

## 10. Conclusions and Prospect

Terrestrial animals and plants have always been important sources of natural products. Humans began to extract active ingredients from terrestrial plants and animals to treat diseases thousands of years ago. With the progress of drug discovery, it is increasingly difficult to develop new molecular entity drugs from terrestrial animals and plants, which cannot cope with the increasing threat to human life and health. Therefore, "asking for drugs from the sea" has become a key to find new drug sources. The continuous innovation of deep-sea mining technology, extraction and separation technology, molecular modification technology, genetic engineering technology, and organic synthesis technology provide hope and new opportunities for the development of marine anticancer drugs. 

## Figures and Tables

**Figure 1 marinedrugs-19-00488-f001:**
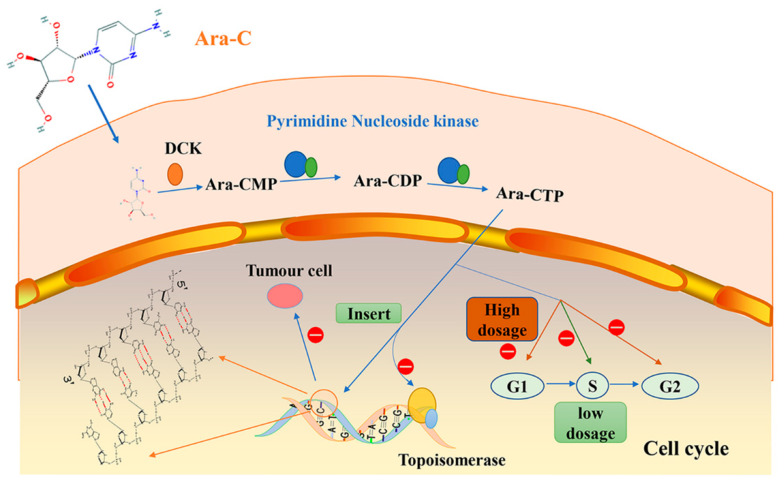
Cytarabine exhibits anticancer effects by inhibiting DNA synthesis and inducing cell cycle arrest.

**Figure 2 marinedrugs-19-00488-f002:**
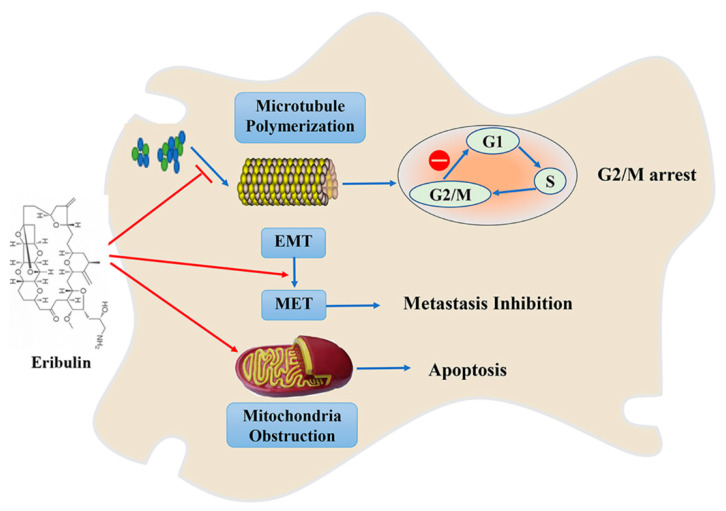
Eribulin exerts its anticancer effects by inducing cell cycle arrest, apoptosis, and tumor metastasis inhibition.

**Figure 3 marinedrugs-19-00488-f003:**
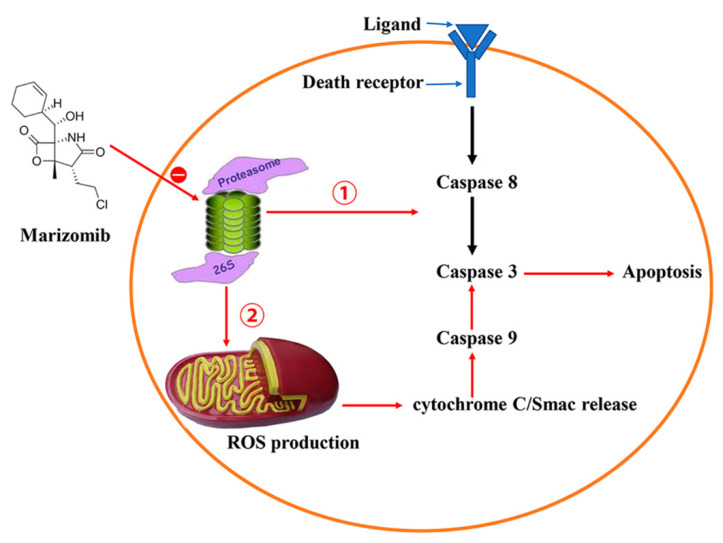
Marizomib exerts its anticancer effects by inducing mitochondria and death receptor pathways.

**Figure 4 marinedrugs-19-00488-f004:**
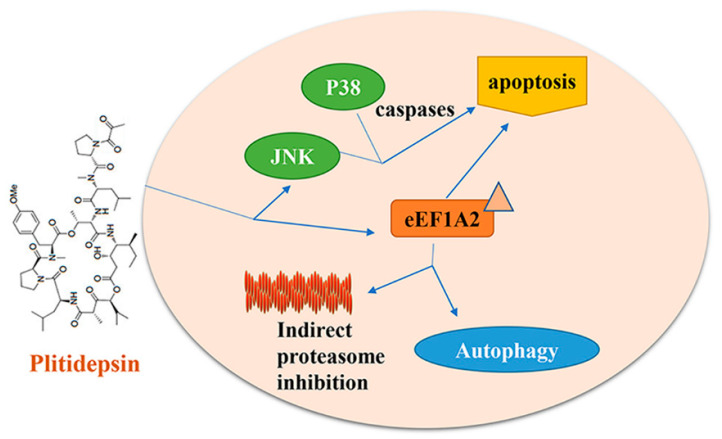
Plitidepsin displays anticancer effects by inducing cell apoptosis and autophagy.

**Figure 5 marinedrugs-19-00488-f005:**
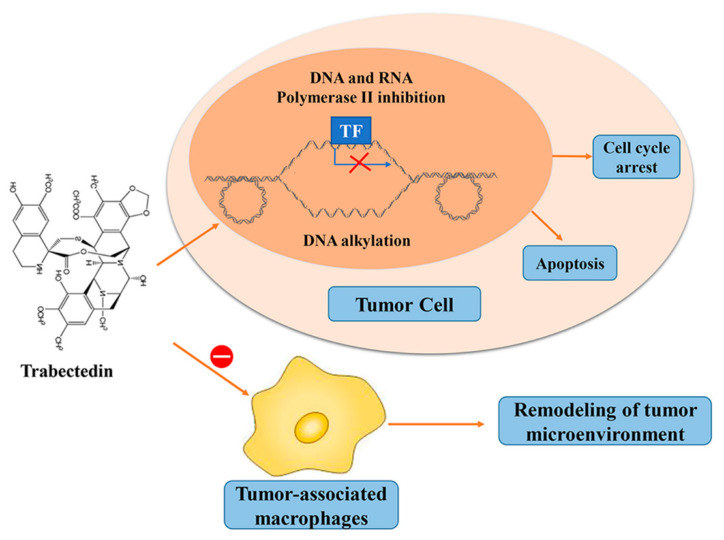
Trabectedin displays anticancer effects through inducing cell cycle arrest, apoptosis, and remodeling of tumor microenvironment. (TF: transcription factor).

**Figure 6 marinedrugs-19-00488-f006:**
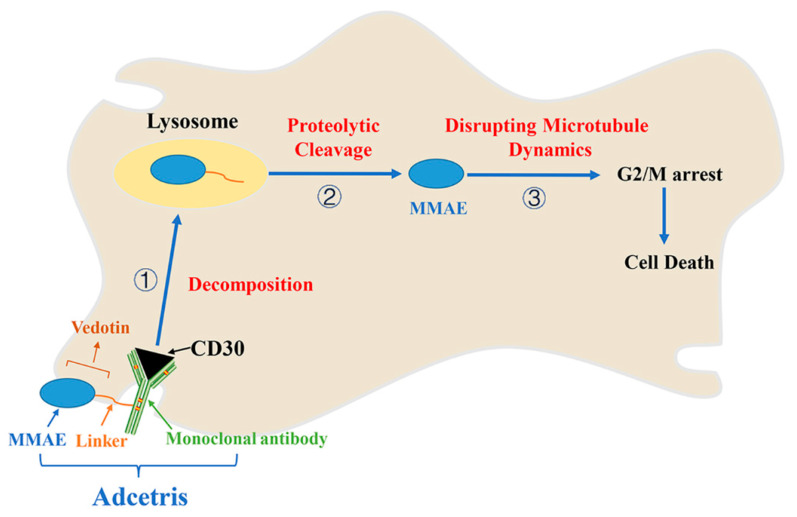
The action model of adcetris targeting CD30 positive tumor cells.

**Figure 7 marinedrugs-19-00488-f007:**
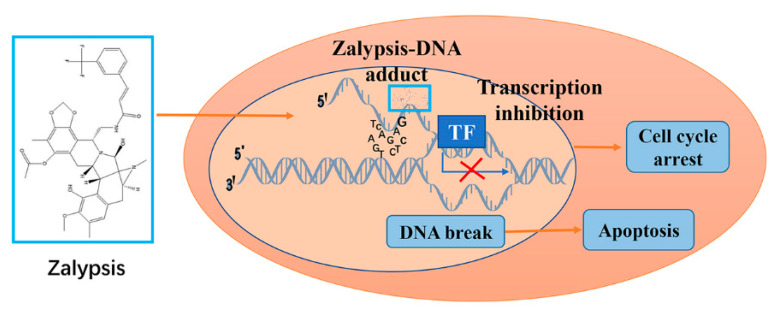
Zalypsis forms a DNA-drug adduct which induces cell cycle arrest and apoptosis.

**Figure 8 marinedrugs-19-00488-f008:**
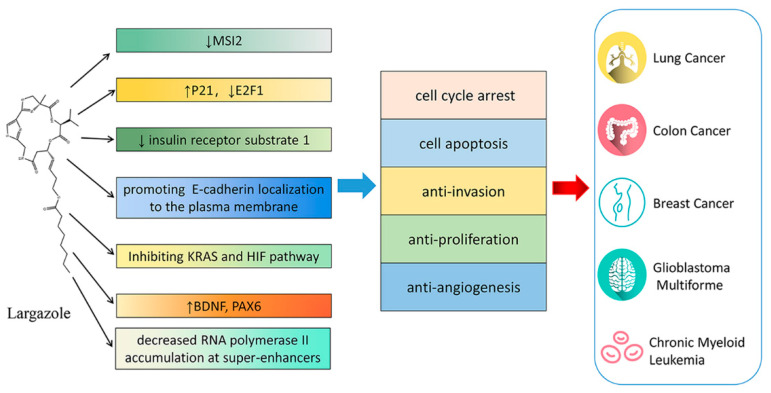
Anticancer mechanism of largazole.

**Figure 9 marinedrugs-19-00488-f009:**
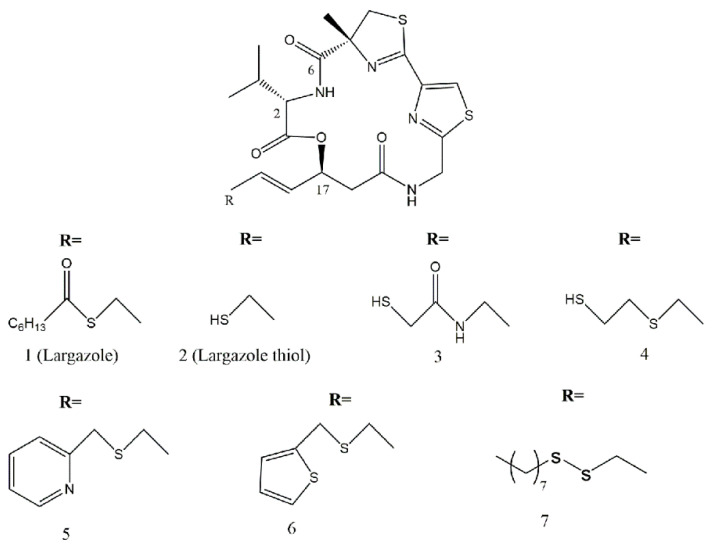
Largazole, largazole thiol, and largazole analogues with different ZBGs.

**Figure 10 marinedrugs-19-00488-f010:**
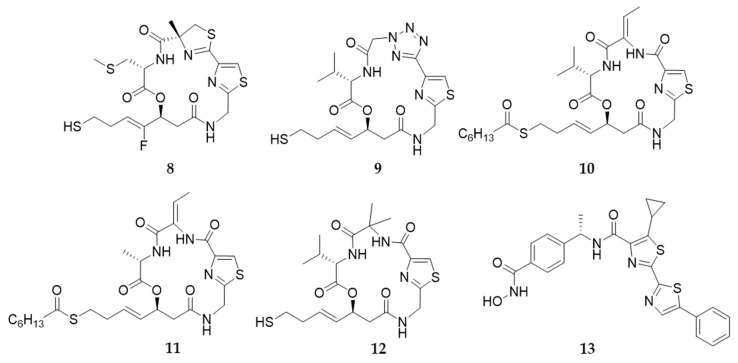
Largazole analogues with a modified depsipeptide ring.

**Table 1 marinedrugs-19-00488-t001:** Selected marine natural compounds or derivatives that have been approved or entered into clinical trials in the past decades.

Compound	Source	Indications	Mechanisms	Clinical Trial
Cytarabine	*Cryptotethia crypta*	AML	DNA synthesis inhibition	Approved
Eribulin	*Halichondria okadai*	Breast cancer, liposarcoma	Prevention the formation of spindles and EMT, apoptosis induction	Approved
Marizomib	*Salinispora tropica*	MM	Selectively inhibiting 20S proteasome	Approved
Plitidepsin	*Aplidium albican*	Solid tumor, MM	Cell cycle arrest, apoptosis induction	Approved
Trabectedin	*Ecteinascidia turbinata*	Liposarcoma, leiomyosarcoma	Binding to guanine residues in the DNA groove to form protein adducts	Approved
Adcetris	*Dolabella auricularia*	Hodgkin’s lymphoma	Inhibiting tubulin formation, apoptosis induction	Approved
Zalypsis	*Joruna funebris*	Urothelial carcinoma, cervical carcinoma, MM	rendering DNA double strand breaks, cell cycle arrest, apoptosis induction	Phase II
OKI-179	*Cyanobacterium, Symploca*	Advanced solid tumors	Class I HDAC inhibition	Phase I

Abbreviations: HDAC: Histone Deacetylase; AML: Acute Myeloid Leukemia; EMT: Epithelial-Mesenchymal Transition; MM: Multiple Myeloma; MMPs: Matrix Metalloproteinases.

## Data Availability

No new data were created or analyzed in this study. Data sharing is not applicable to this article.

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
