# Peer review of "Marine Power on Cancer: Drugs, Lead Compounds, and Mechanisms"

_marinedrugs, 2021, doi:10.3390/md19090488_

Round 1
Reviewer 1 Report
Reviews about new pharmaceuticals are an ever-green, due to their great importance.
It is always interesting to read success stories, it is really stimulating.
However, I am missing chemical structures for most chemical entities described. Only cytarabine is shown in figure 1, and several largazole analoges in figures 7 and 8. As a chemist I would like to see chemical structures for all the new drugs described. Figures 2-4 can be shrunk instead.
Author Response
Reviews about new pharmaceuticals are an ever-green, due to their great importance. It is always interesting to read success stories, it is really stimulating. However, I am missing chemical structures for most chemical entities described. Only cytarabine is shown in figure 1, and several largazole analoges in figures 7 and 8. As a chemist I would like to see chemical structures for all the new drugs described. Figures 2-4 can be shrunk instead.
-- Many thanks to the reviewer for the comment. As suggested the chemical structures of the drugs have been displayed in revised figures (Figures 1 to 10) which were shrunk.
Reviewer 2 Report
The article entitled "Marine power on cancer: drugs, lead compounds, and mechanism" describes some of the commonest marine-derived drugs used in the treatment of both solid and hematological malignancies, deepening origin, mechanism of action and clinical use. Examples of their activity in in vitro and in vivo models are further reported, as well as new analogues with interesting results and perspectives as marine anti-cancer agents.
Although not all the currently approved marine-derived compounds are discussed, those selected for the review make the article interesting and suitable, in my opinion, for publishing to Marine drugs after minor revisions.
I suggest to the authors to insert in figure 2, 3, 4 and 5 the chemical structure of the related drugs, instead of geometric shapes. Moreover, unlike the other drugs, for Marizomib and Zalypsis no figures summarizing the mechanism of action are reported. In order to harmonise the paragraphs, they should be included.
- Please check that in vitro, in vivo and the marine species of origin are written in italics;
- At page 1, lines 12-13, remove “there are an estimated” and correct “occurred” with the present form;
- At page 1, lines 22-25, the sentence “The future development of anti-cancer..” is not clear. Please rephrase or delate it;
- At page 1, line 28, insert “environment” after “marine”;
- At page 1, line 34, insert the following reference after “have been developed” to give an overview on all the marine anticancer agents approved so far: Mar. Drugs 2020, 18, 619; Eur. J. Med. Chem. (2020), 204, 112631 ; J. Med. Chem. (2020), 63(20), 12023-12042; J. Med. Chem. (2016), 59, 7223.
- At page 1, line 39, change “largazole” with “OKI-179” since only the latter is reported in table 1;
- In table 1, Adcetris is listed as in clinical development (phase I) but it has been approved in 2011 by FDA for the treatment of HL;
- At page 2, line 46, insert “sponge” after “marine”;
- At page 2, line 54, correct “20β” with “2’-β”
- At page 2, line 58, correct “procession” with “process”;
- At page 4, line 107, insert “type of” after “various”;
- At page 4, line 133, substitute the second “lactam” with “β-lactone”;
- At page 4, line 134, remove “class of”;
- At page 5, line 161, remove “marine animal” and insert the exact animal species;
- At page 5, line 165, change “mainly elaborated” with “deeply investigated”;
- At page 5, line 178, reposition “indirectly” in the sentence;
- At page 6, line 189, change “tetrahydro isoquinoline” with “tetrahydroisoquinoline”;
- In paragraph 5, DNA is often written as “deoxyribonucleic acid”. Please uniform;
- At page 6, line 202, remove “that”;
- At page 7, lines 223-226, the last sentence is not clear. Please rephrase it;
- At page 8, lines 243 and 245, change “was” with “is”;
- At page 8, line 247, correct the acronym of anaplastic large cell lymphoma. It is ALCL, not HL;
- At page 8, better describe the structure of brentuximab vedotin, highlighting spacer – peptide cleavable linker and attachment group;
- Since brentuximab vedotin is currently under evaluation in more than 80 active phase I-III clinical trials, a comment should be made about it; in this contest, insert the following references: Pharmacology & Therapeutics 211 (2020) 107552;
- At page 9, line 279, correct “HADCs” with “HDACs”;
- At page 9, line 288, avoid the repetition of “Luesch and coworkers”;
- At page 9, line 292, rephrase “a large body of literature”;
- At page 9, line 311, correct “decreased” with “decrease”;
- At page 11, the compounds displayed in figure 7 can be reunited in a single chemical structure, distinguishing the various ZBGs in a table or caption;
- At page 12, line 395, change “in mechanism” with “according to the mechanism of action”;
- Check the style of references 45 and 50.
Author Response
- I suggest to the authors to insert in figure 2, 3, 4 and 5 the chemical structure of the related drugs, instead of geometric shapes. Moreover, unlike the other drugs, for Marizomib and Zalypsis no figures summarizing the mechanism of action are reported. In order to harmonise the paragraphs, they should be included.
-- We thank the reviewer for the kind comment. As suggested, the chemical structures of the drugs have been displayed in revised figures, and the anti-cancer mechanisms of action of marizomib and zalypsis were depicted in Figure 3 and Figure 7.
- Please check that in vitro, in vivo and the marine species of origin are written in italics;
-- Many thanks. We have carefully checked through the manuscript and made the revisions on line 17, 59, 190, 233-234, 246, 300, 317, 327, and 404, so that in vitro, in vivo and the marine species of origin are written in italics.
- At page 1, lines 12-13, remove “there are an estimated” and correct “occurred” with the present form;
-- The description of “there are an estimated” has been deleted and the past tense has been corrected as the present form in line 12 highlighted in red.
- At page 1, lines 22-25, the sentence “The future development of anti-cancer..” is not clear. Please rephrase or delate it;
-- “the forward and reverse” has been replaced by “the innovation” in line 23, thank you.
- At page 1, line 28, insert “environment” after “marine”;
-- The word “environment” has been inserted after “marine” in line 26.
- At page 1, line 34, insert the following reference after “have been developed” to give an overview on all the marine anticancer agents approved so far: Mar. Drugs 2020, 18, 619; Eur. J. Med. Chem. (2020), 204, 112631 ; J. Med. Chem. (2020), 63(20), 12023-12042; J. Med. Chem. (2016), 59, 7223.
-- These four references have been cited as references 7-10.
- At page 1, line 39, change “largazole” with “OKI-179” since only the latter is reported in table 1;
-- We thank the reviewer for the comment. OKI-179 is an analogue of largazole, so OKI-179 is changed to “largazole analogue OKI-179” in line 40.
- In table 1, Adcetris is listed as in clinical development (phase I) but it has been approved in 2011 by FDA for the treatment of HL;
-- We thank the reviewer for the comment, and revised the Table 1 accordingly.
- At page 2, line 46, insert “sponge” after “marine”;
-- The word “sponge” has been inserted after marine in line 48.
- At page 2, line 54, correct “20β” with “2’-β”
-- Thanks. The word has been corrected accordingly.
- At page 2, line 58, correct “procession” with “process”;
-- Thanks. The word has been corrected accordingly.
- At page 4, line 107, insert “type of” after “various”;
-- The words “type of” has been inserted, thanks.
- At page 4, line 133, substitute the second “lactam” with “β-lactone”;
-- The second “lactam” has been replaced with “β-lactone” in line 134, thank you.
- At page 4, line 134, remove “class of”;
-- The word “class of” has been removed accordingly, thanks.
- At page 5, line 161, remove “marine animal” and insert the exact animal species;
-- We thank the reviewer and deleted the word “marine animal” accordingly.
- At page 5, line 165, change “mainly elaborated” with “deeply investigated”;
-- We thank the reviewer and revised the words accordingly.
- At page 5, line 178, reposition “indirectly” in the sentence;
-- The sentence has been changed to “In chronic lymphocytic leukemia (CLL), plitidepsin affects the viability of leukemic cells through direct and indirect pathways. It inhibits the malignant B-CLL clone directly and restrains tumor growth by modifying the microenvironment indirectly”. Thanks.
- At page 6, line 189, change “tetrahydro isoquinoline” with “tetrahydroisoquinoline”;
-- The word “tetrahydro isoquinoline” has been changed with “tetrahydroisoquinoline” in line 192, thanks.
- In paragraph 5, DNA is often written as “deoxyribonucleic acid”. Please uniform;
-- DNA has been indicated that it is the abbreviation of “deoxyribonucleic acid” in line 194, and thereafter used as DNA. Thanks.
- At page 6, line 202, remove “that”;
-- The phrase “which is a process dependent on transcription factors that called transcription coupling Nucleotide excision repair” has been removed. Thanks.
- At page 7, lines 223-226, the last sentence is not clear. Please rephrase it;
-- The last sentence has been changed to “Moreover, trabectedin induced DNA damage in an RNA-DNA hybrid-dependent manner in Hela cells. This RNA-DNA hybrid is also called R-loop, and high level of R-loops sensitizes tumor cells to trabectedin”, thanks.
- At page 8, lines 243 and 245, change “was” with “is”;
-- We thank the reviewer and made corrections accordingly.
- At page 8, line 247, correct the acronym of anaplastic large cell lymphoma. It is ALCL, not HL;
-- We thank the reviewer and made corrections accordingly.
- At page 8, better describe the structure of brentuximab vedotin, highlighting spacer – peptide cleavable linker and attachment group;
-- Thank you. “To meet the requirements, the main ingredient of linker in adcetris is cathepsin cleavable valine‐citrulline, which is able to be cut accurately to release cytotoxic compound” has been added (page 7, lines 243-245.
- Since brentuximab vedotin is currently under evaluation in more than 80 active phase I-III clinical trials, a comment should be made about it; in this contest, insert the following references: Pharmacology & Therapeutics 211 (2020) 107552;
-- We thank the reviewer for the suggestion, and included a comment in lines 262-270 highlighted. Also, the mentioned paper has been cited as reference 91.
- At page 9, line 279, correct “HADCs” with “HDACs”;
-- We thank the reviewer and made corrections accordingly.
- At page 9, line 288, avoid the repetition of “Luesch and coworkers”;
-- Thank you for the comment. The second “Luesch and coworkers” has been replaced by “The research group” in line 301.
- At page 9, line 292, rephrase “a large body of literature”;
-- Thank you for the suggestion. “a large body of literature” has been replaced by “Studies” in line 305.
- At page 9, line 311, correct “decreased” with “decrease”;
-- We thank the reviewer and made corrections accordingly.
- At page 11, the compounds displayed in figure 7 can be reunited in a single chemical structure, distinguishing the various ZBGs in a table or caption;
-- As suggested, the former figure 7 has been changed to Figure 9 on page 11. Figure 10 shows largazole analogues with a modified depsipeptide ring.
- At page 12, line 395, change “in mechanism” with “according to the mechanism of action”;
-- We thank the reviewer and made revisions accordingly.
- Check the style of references 45 and 50.
-- We thank the reviewer and made revisions accordingly.
Reviewer 3 Report
The authors review marine natural products as promising anticancer agents. Several reviews have already been published on this topic like Mar Drugs. 2019 Sep; 17(9): 491, Mar. Drugs 2020, 18(12), 619. The authors selected 8 anticancer agents and 5 of them are on the market the others being in phase I or II of clinical trials. The main interest of the review is on the nicely illustrated mechanisms of action of the selected marine drugs. They should clarify in which aspect their review is original.
The authors should pay much more attention to the description of the origin of the compounds: sponge, ascidian etc. The statements are frequently incorrect and they should be revised. We can regret not seeing the structures of the selected drugs. The English is quite poor and should be revised thoroughly as it is sometimes very difficult to read/understand
In the conclusion, the authors mention a depletion of terrestrial resources. It seems a bit too much as species are also in danger in our oceans
Author Response
- The authors review marine natural products as promising anticancer agents. Several reviews have already been published on this topic like Mar Drugs. 2019 Sep; 17(9): 491, Mar. Drugs 2020, 18(12), 619. The authors selected 8 anticancer agents and 5 of them are on the market the others being in phase I or II of clinical trials. The main interest of the review is on the nicely illustrated mechanisms of action of the selected marine drugs. They should clarify in which aspect their review is original.
-- We thank the reviewer for the comment. In the present review, the original resource, anti-cancer effects, especially the mechanisms of action and clinical applications of selected drugs were mainly discussed. Thank you.
- The authors should pay much more attention to the description of the origin of the compounds: sponge, ascidian etc. The statements are frequently incorrect and they should be revised.
--We thank the reviewer for the comment. We have carefully checked through the manuscript and made the corrections on line 17, 59, 190, 233-234, 246, 300, 317, 327, and 404. Thank you.
- We can regret not seeing the structures of the selected drugs.
-- The chemical structures of the drugs have been displayed in revised figures, thank you.
- The English is quite poor and should be revised thoroughly as it is sometimes very difficult to read/understand.
-- We have carefully revised the manuscript in order to make it easy to be understood. Thank you.
- In the conclusion, the authors mention a depletion of terrestrial resources. It seems a bit too much as species are also in danger in our oceans.
-- We thank the reviewer for the comment. In the revised version, the original words “With the increasingly depletion of terrestrial resources” has been deleted.
Round 2
Reviewer 3 Report
Some efforts have been made to improve this review. The English still needs significant improvements. For example the first sentence:
Marine has been a huge treasure of medicinal resources
I would suggest the authors find a native speaker or a service for the English review.
Author Response
The manuscript has been edited by natives speaking English. Thanks.